# Prognostic Role of TAPSE to PASP Ratio in Patients Undergoing MitraClip Procedure

**DOI:** 10.3390/jcm10051006

**Published:** 2021-03-02

**Authors:** Blanca Trejo-Velasco, Rodrigo Estevez-Loureiro, Fernando Carrasco-Chinchilla, Felipe Fernández-Vázquez, Dabit Arzamendi, Manuel Pan, Isaac Pascual, Luis Nombela-Franco, Ignacio J. Amat-Santos, Xavier Freixa, Rosa Ana Hernández-Antolín, Ramiro Trillo-Nouche, Leire Andraka Ikazuriaga, José R. López-Mínguez, Dario Sanmiguel Cervera, Juan Sanchis, José Luis Diez-Gil, Valeriano Ruiz-Quevedo, Cristóbal Urbano-Carrillo, Víctor Manuel Becerra-Muñoz, Tomás Benito-González, Chi Hion Li, Dolores Mesa, Pablo Avanzas, Germán Armijo, Ana María Serrador-Frutos, Laura Sanchis, Covadonga Fernández-Golfín Lobán, Belén Cid-Álvarez, José María Hernández-García, Carmen Garrote-Coloma, Estefanía Fernández-Peregrina, Miguel Romero, Víctor León Arguero, Ignacio Cruz-González

**Affiliations:** 1Cardiology Department, University Hospital of Salamanca, Biomedical Research Institute of Salamanca (IBSAL), 37007 Salamanca, Spain; cruzgonzalez.ignacio@gmail.com; 2Cardiology Department, Alvaro Cunqueiro University Hospital, 36213 Vigo, Spain; roiestevez@hotmail.com; 3Cardiology Department, Virgen de la Victoria University Hospital, Biomedical Research Institute of Málaga, 29010 Málaga, Spain; fernandocarrascochinchilla@gmail.com (F.C.-C.); vmbecerram@gmail.com (V.M.B.-M.); josemaria2509@gmail.com (J.M.H.-G.); 4Biomedical Research Networking Center on Cardiovascular Diseases (CIBERCV), 28029 Madrid, Spain; ijamat@gmail.com (I.J.A.-S.); ramirotrillo@mac.com (R.T.-N.); sanchis_juafor@gva.es (J.S.); belcid77@hotmail.com (B.C.-Á.); 5Cardiology Department, University Hospital of Leon, 24071 Leon, Spain; ffernandez@secardiologia.es (F.F.-V.); tomasbenito@outlook.com (T.B.-G.); cgarrote@saludcastillayleon.es (C.G.-C.); 6Cardiology Department, University Hospital Santa Creu i Sant Pau, 08041 Barcelona, Spain; dabitarza@gmail.com (D.A.); ch.pedroli@gmail.com (C.H.L.); EFernandezPe@santpau.cat (E.F.-P.); 7Cardiology Department, University Hospital Reina Sofia, University of Cordoba and Maimonides Institute for Biomedical Research (IMIBIC), 14004 Córdoba, Spain; manuelpanalvarez@gmail.com (M.P.); loladoctora@gmail.com (D.M.); mromero@grupocorpal.com (M.R.); 8Heart Area, Asturias Central University Hospital, University of Oviedo, Health Research Institute, 33011 Oviedo, Spain; ipascua@live.com (I.P.); avanzas@gmail.com (P.A.); leonarguero@yahoo.es (V.L.A.); 9Cardiovascular Institute, Hospital Clínico San Carlos, Health Research Institute San Carlos (IdISSC), 28040 Madrid, Spain; luisnombela@yahoo.com (L.N.-F.); garmijo.md@gmail.com (G.A.); 10Cardiology Department, Clinic University Hospital of Valladolid, Heart Sciences Institute, 47003 Valladolid, Spain; aserradorf@gmail.com; 11Cardiology Department, Clínic University Hospital of Barcelona, University of Barcelona, Biomedical Research Institute August Pi i Sunyer (IDIBAPS), 08041 Barcelona, Spain; xavierfreixa@hotmail.com (X.F.); lsanchis@clinic.cat (L.S.); 12Cardiology Department, Ramon y Cajal University Hospital, 28034 Madrid, Spain; rhernandez_antolin@hotmail.com (R.A.H.-A.); covagolfin@yahoo.es (C.F.-G.L.); 13Cardiology Department, Clinical University Hospital of Santiago, 15706 Santiago de Compostela, Spain; 14Cardiology Department, University Hospital of Basurto, 48013 Bilbao, Spain; Leire.andrakaikazuriaga@osakidetza.eus; 15Cardiology Department, University Hospital of Badajoz, 06080 Badajoz, Spain; lopez-minguez@hotmail.com; 16Cardiology Department, General University Hospital of Valencia, 46014 Valencia, Spain; dasancer@hotmail.com; 17Cardiology Department, University Clinic Hospital of Valencia, University of Valencia, 46010 Valencia, Spain; 18Cardiology Department, University & Polytechnical Hospital La Fe, 46026 Valencia, Spain; diez_jlu@gva.es; 19Cardiology Department, University Hospital of Navarra, 31008 Pamplona, Spain; valeriano.ruiz.quevedo@navarra.es; 20Cardiology Department, Regional University Hospital of Málaga, 29010 Málaga, Spain; cristobalurbano@gmail.com; 21Cardiology Department, University of Salamanca, 37008 Salamanca, Spain

**Keywords:** MitraClip, mitral valve repair, mitral valve regurgitation, pulmonary hypertension, right ventricular to pulmonary arterial coupling, transthoracic echocardiography

## Abstract

Background: Transcatheter mitral valve repair (TMVR) is an effective therapy for high-risk patients with severe mitral regurgitation (MR) but heart failure (HF) readmissions and death remain substantial on mid-term follow-up. Recently, right ventricular (RV) to pulmonary arterial (PA) coupling has emerged as a relevant prognostic predictor in HF. In this study, we aimed to assess the prognostic value of tricuspid annular plane systolic excursion (TAPSE) to PA systolic pressure (PASP) ratio as a non-invasive measure of RV-to-PA coupling in patients undergoing TMVR with MitraClip (Abbott, CA, USA). Methods: Multicentre registry including 228 consecutive patients that underwent successful TMVR with MitraClip. The sample was divided in two groups according to TAPSE/PASP median value: 0.35. The primary combined endpoint encompassed HF readmissions and all-cause mortality. Results: Mean age was 72.5 ± 11.5 years and 154 (67.5%) patients were male. HF readmissions and all-cause mortality were more frequent in patients with TAPSE/PASP ≤ 0.35: Log-Rank 8.844, *p* = 0.003. On Cox regression, TAPSE/PASP emerged as a prognostic predictor of the primary combined endpoint, together with STS-Score. TAPSE/PASP was a better prognostic predictor than either TAPSE or PASP separately. Conclusions: TAPSE/PASP ratio appears as a novel prognostic predictor in patients undergoing MitraClip implantation that might improve risk stratification and candidate selection.

## 1. Introduction

Over the past decade, transcatheter mitral valve repair (TMVR) with the MitraClip system (Abbott; Menlo Park, California, USA) has proven a safe and effective treatment alternative for high-risk patients with severe mitral regurgitation (MR) [1,2,3,4].

Currently, the indication for TMVR with the MitraClip system as an alternative to surgery is set by a dedicated Heart Team, after a thorough evaluation of each individual case, including appropriateness and feasibility of the procedure, risk stratification and benefit prediction [5,6]. This decision-making process is supported by surgical risk models, which have, however, exhibited limited accuracy in patients undergoing TMVR with MitraClip [2,7]. In addition, relevant prognostic indicators identified by prior studies in this population are also taken into account [8,9,10,11]. Notwithstanding, heart failure (HF) readmissions and death during the first year after MitraClip implantation remain substantial [2,3,8,9,10,11], even after a successful procedure. Thus, refinement of current prognostic predictors to improve candidate selection and avoid futility remains an important unanswered question.

Patients with severe MR considered for a MitraClip procedure often associate pulmonary hypertension (PH) and right ventricular (RV) dysfunction, which are indicators of a worse outcome [12,13,14]. Indeed, many patients are declined a MitraClip procedure due to PH and RV dysfunction, but TMVR may still provide clinical benefit in selected patients and has been associated with a significant reduction in pulmonary pressures and RV reverse remodelling in several studies [14,15,16].

Recently, RV to pulmonary arterial (PA) coupling has emerged as a relevant prognostic indicator in HF and PH [17,18,19]. RV-PA coupling quantifies the adaptation of the RV to its afterload and has the ability to detect pending RV failure [17,18,19,20]. In clinical practice, RV to PA coupling is frequently assessed non-invasively by means of the ratio between tricuspid annular plane systolic excursion (TAPSE) and pulmonary artery systolic pressure (PASP) [19,20,21,22]. However, the prognostic role of TAPSE/PASP ratio in patients undergoing MitraClip implantation has not been evaluated to date.

In this study we aimed to assess the ability of baseline TAPSE/PASP ratio to improve prognostic stratification in patients undergoing successful TMVR with MitraClip.

## 2. Materials and Methods

### 2.1. Study Design and Population

228 consecutive patients undergoing TMVR with MitraClip between June 2012 and June 2018 at 17 Spanish centres were included.

Patients’ data were obtained from the Spanish MitraClip observational multicentre Registry, endorsed by the Cardiac Catheterization and Interventional Cardiology Section of the Spanish Society of Cardiology. The sample included patients with primary and secondary MR graded as moderate to severe (3+) or severe (4+) on pre-procedural echocardiography. Only patients that displayed persistent HF symptoms despite optimized guideline directed medical therapy were included. At each of the centres, the corresponding Heart Team established the indication of TMVR with MitraClip as the best treatment alternative, after careful evaluation of each case. Follow-up data were prospectively collected at each participating center by serial clinical, analytical and echocardiographic evaluations. Only patients with a comprehensive baseline echocardiographic examination that underwent successful TMVR with residual MR grade ≤ 2+ were included in the current analysis.

The Ethics Committee of each of the participating centres approved the Study Protocol and all patients provided written informed consent prior to their enrolment.

### 2.2. Echocardiography Parameters

MR severity was graded as none, mild (1+), moderate (2+), moderate to severe (3+) and severe (4+) according to regurgitant volume and effective regurgitant orifice area. TAPSE was measured as the peak excursion of the tricuspid annulus from end-diastole to end-systole in the apical 4-chamber view with the M-mode. RV systolic pressure was determined from the peak velocity of the tricuspid regurgitation (TR) jet using the simplified Bernoulli equation. Right atrial pressure was estimated by the diameter and collapsibility of the inferior vena cava and added to the calculated gradient to yield PASP. Echocardiography measurements were recorded and averaged over three consecutive heart cycles in patients in sinus rhythm and over 3–5 heart cycles in patients in atrial fibrillation.

### 2.3. Right Heart Catheterization

Right heart catheterization (RHC) was performed via femoral or brachial vein employing standard methodology in a small subset of the total sample, at the discretion of the patient’s physician and standard hemodynamic measurements were registered.

### 2.4. Definitions and Outcomes

The details of the mitral repair procedure have been previously described [1]. All procedures were guided by transesophagic echocardiography. Procedural success was defined as a reduction in the degree of MR equal to or less than moderate ≤ 2+. Functional class was defined according to the New York Heart Association (NYHA) classification and was assessed at baseline and during follow-up.

The primary combined endpoint was defined as the composite of HF readmissions and all-cause death during the first-year post-procedure, including in-hospital mortality. Secondary endpoints consisted of the occurrence of each of these events separately. Follow-up for HF hospitalizations began after discharge from index admission.

Periprocedural complications and in-hospital and thirty-day outcomes were defined as per Mitral Valve Academic Research Consortium criteria [23] and encompassed pericardial effusion, air embolism, cordal rupture, catheter thrombosis, clip detachment, bleeding assessed according to Bleeding Academic Research Consortium (BARC) criteria, access-site vascular complications, myocardial infarction, acute kidney injury, pulmonary embolism, stroke, in-hospital mortality and 30-day readmission for decompensated HF.

### 2.5. Statistical Analysis

Categorical variables were expressed as frequencies and percentages and continuous variables as mean±standard deviation. The sample was divided in two groups according to the median value of baseline TAPSE/PAPS ratio.

Comparisons between patient’s groups according to TAPSE/PASP ratio or event status were made by *χ*^2^ test, unpaired Student’s *t*-test and Mann–Whitney U-test, as appropriate.

Survival curves for baseline TAPSE/PAPS were constructed with the Kaplan–Meier method and compared with the Log-Rank test. A multivariate Cox regression model was performed, including all variables with a *p*-value ≤ 0.10 on univariate analysis as well as variables expected to influence outcome based on previous publications, after exclusion of colinearity. Discrimination measures by means of Harrell’s C-statistics were calculated for Cox regression model and the proportional hazard assumption for exposure of variables included in the model was verified.

Correlation analysis was performed by means of Pearson correlation coefficient to assess for the existence of a linear relationship between baseline TAPSE/PASP ratio and other hemodynamic parameters.

A *p*-value < 0.05 was considered statistically significant for all tests. Statistical analyses were performed with SPSS 21.0 (IBM, Chicago, IL, US) for Windows.

## 3. Results

A total of 228 consecutive patients were included in our study. Mean age was 72.5 ± 11.5 years and 154 (67.5%) patients were male. The vast majority of patients were highly symptomatic at baseline (NYHA functional class ≥III/IV, 86.8%) and surgical risk was moderately increased (mean STS 5.8 ± 5.3, mean EuroSCORE-II 8.5 ± 7.9). MR was severe in 187 (82.1%) patients and moderate-severe in the remainder 41 (17.9%) cases. The underlying MR aetiology was classified as secondary in 147/225 (65.3%) patients, primary in 50/225 (22.2%) cases, mixed aetiology in 28/225 (12.3%) and was not reported in 3 cases. All patients underwent successful TMVR with the MitraClip device with 1.5 ± 0.6 clips on average. Postprocedural MR was reduced to grade 0–1+ in 147 (64.5%) patients and to grade 2+ in the rest.

### 3.1. Baseline Characteristics According to TAPSE/PASP Values

Median value of TAPSE/PAPS was 0.35, with lower values indicating RV to PA uncoupling, in agreement with cutoff values with prognostic implications reported in prior studies in HF [19,20,24,25]. Patients with TAPSE/PASP ratio ≤0.35 presented a higher prevalence of grade 3–4+ TR, lower TAPSE, greater PASP and higher left atrial (LA) v-wave and PASP values determined by RHC, Table 1. No further differences in baseline characteristics according to TAPSE/PASP ratio existed.

Next, the correlation between TAPSE/PASP ratio, a non-invasive index of the RV to PA coupling state and RHC derived variables, including RV to PA coupling parameters, were assessed by means of the Pearson correlation coefficient. TAPSE/PASP presented a positive linear relationship with PA compliance and a negative linear relationship with PA pulse pressure, PASP and LA v-wave pressure, Table 2.

### 3.2. Outcomes

During a median follow-up period of 260 days (IQ range: 115–394), the primary combined endpoint occurred in 71 (31.1%) patients (40.4% TAPSE/PASP ≤ 0.35 vs. 21.9% in TAPSE/PASP > 0.35, *p* = 0.003), at the expense of 61 (26.8%) HF admissions and 22 (9.6%) all-cause deaths. 12 deaths occurred in patients with a previous hospitalization for HF and 2 during index hospital admission. Cardiovascular mortality accounted for 10 (45.5%) of total deceases.

Higher rates of the primary combined endpoint were noted among patients with diabetes (45.1% vs. 26.8%, *p* = 0.006) and NYHA III to IV functional class at baseline (94.4% vs. 83.4%, *p* = 0.024). STS-Score was numerically higher in patients that met the primary endpoint, although this association was not statistically significant (7.1 ± 6.4 vs. 5.3 ± 4.7, *p* = 0.059), (Appendix A). Moreover, the primary combined endpoint occurred more frequently among patients with lower TAPSE and TAPSE/PASP ratio, but there were no significant differences in PASP values. Incidence of device-related and periprocedural complications were low across the whole sample and did not differ according to the primary combined endpoint, with the exception of in-hospital mortality and 30-day readmissions for decompensated HF, (Appendix A).

Importantly, a more advanced NYHA III-IV/IV functional class during follow-up was more frequent in patients with reduced TAPSE/PASP at baseline (39.5% vs. 25.7%, *p* = 0.045) and also among patients that maintained a reduced TAPSE/PASP ≤ 0.35 after successful TMVR (55.9% vs. 27.8%, *p* = 0.004). Of note, TAPSE/PASP ratio raised over the 0.35 threshold in almost two thirds of patients with baseline TAPSE/PASP ≤ 0.35 that had this parameter reassessed after TMVR, (Appendix A).

### 3.3. Survival and Multivariate Analysis

On Kaplan–Meier survival analysis, patients with TAPSE/PASP ≤ 0.35 had a higher risk of suffering the primary combined endpoint, Log Rank 8.844, *p* = 0.003, Figure 1a. When cardiac events were analysed separately, HF readmissions (Log Rank 7.810, *p* = 0.005; Figure 1b) remained significantly more frequent among patients with TAPSE/PASP ≤ 0.35, but there were no differences regarding global mortality according to TAPSE/PASP (Log-Rank 0.002, *p* = 0.962; Figure 1c).

In order to assess whether TAPSE/PASP acted as an independent prognostic marker, a Cox proportional hazard analysis was performed, including all baseline and periprocedural characteristics that presented a *p*-value < 0.10 on univariate analysis. The ability of the TAPSE/PASP ratio to predict the combined endpoint was assessed against that of TAPSE and PASP to avoid colinearity, and, as only TAPSE/PASP maintained its prognostic significance, TAPSE and PASP were discarded from the final model.

Multivariate Cox regression identified TAPSE/PASP as an independent prognostic marker for the primary combined endpoint (HR 2.0; 95% CI: 1.13–3.53, *p* = 0.017), alongside with STS-Score (HR 1.05; 95% CI: 1.01–1.10, *p* = 0.024), Table 3.

Interestingly, TAPSE/PASP ratio maintained its prognostic value regardless of MR aetiology. On Kaplan–Meier survival analysis, a TAPSE/PASP ratio ≤ 0.35 was associated with a higher risk for the primary combined endpoint in patients with primary MR, Figure 2a, secondary MR, Figure 2b, and mixed aetiology MR, Figure 2c, Log Rank 8.749, *p* = 0.003. Moreover, when MR aetiology was introduced into Cox Multivariate Regression analysis, TAPSE/PASP ≤ 0.35 remained an independent prognostic indicator for the primary combined endpoint (HR: 2.19, 95% CI: 1.18–3.85, *p* = 0.012), together with STS-Score (HR: 1.05, 95% CI: 1.01–1.10, *p* = 0.021).

## 4. Discussion

To the best of our knowledge, this is the first study to assess the prognostic value of TAPSE/PASP ratio in patients undergoing successful TMVR with the MitraClip system. In this sample, TAPSE/PASP emerged as a prognostic predictor for the primary combined endpoint encompassing HF readmissions and all-cause mortality, alongside with STS-Risk Score. Importantly, TAPSE/PASP ratio provided a better prognostic stratification on multivariate Cox regression analysis than either TAPSE or PASP separately.

RV to PA coupling has recently been established as a valuable prognostic predictor in HF [19,20,21]. The gold standard for determining RV-PA coupling consists of the ratio between RV end-systolic elastance and effective arterial elastance [Ees/Ea] [17,26], which requires invasive measurements by means of a RHC as well as specific, dedicated material and is thus, rarely employed in clinical practice. Recently, the TAPSE/PASP ratio has been proposed as a non-invasive indicator of RV to PA coupling [19,20,21,22,27]. This index provides a valuable non-invasive measure of RV contractile state and RV load adaptability beyond the information afforded by each separate variable as an index of changes in RV length (TAPSE) versus developed force (PASP) [19,21]. Accordingly, TAPSE/PASP ratio may help identify initial stages of RV functional reserve reduction [26], which could account for its improved prognostic accuracy.

Our results are consistent with previous studies that have recently reported the prognostic value of TAPSE/PASP ratio in several HF and PH populations, [19,20,21,22], including patients with severe aortic stenosis undergoing transcatheter aortic valve implantation [28]. Indeed, in our sample, TAPSE/PASP ratio displayed a significant linear correlation with PA compliance and PA pulse pressure, both of which are invasive indicators of the distensibility of the pulmonary vascular bed. This observation provides further support on the value of TAPSE/PASP ratio as a non-invasive index of the RV to PA coupling.

In addition, TAPSE/PASP ratio presented a significant negative correlation with baseline LA v-wave pressure. This finding could reflect greater LA stiffness in patients with worse RV to PA coupling as per TAPSE/PASP values, in agreement with prior studies that link loss of LA reservoir function with RV–PA uncoupling [24] and impaired functional class [29,30]. In fact, a reduced TAPSE/PASP ratio ≤ 0.35, either at baseline or after successful TMVR, was associated with worse functional class during follow-up in our study.

Altogether, identification of readily available, accurate prognostic predictors remains especially relevant in patients undergoing MitraClip implantation to avoid futility, as patients considered for this intervention often present advanced HF and a high load of comorbidities. Prior studies have recognized numerous indicators of worse outcomes in this population, including reduced bi-ventricular function, enlarged left ventricular (LV) volumes, PH, severe TR and advanced functional class, amongst others [3,9,10,11,12,13,14,15,31].

Our study adds to aforementioned trials as it identifies a novel parameter, the TAPSE/PASP ratio, which improves prognostic stratification of patients undergoing TMVR adding to already known risk indicators, regardless of MR aetiology. This is finding is important considering the contradictory results on the utility of MitraClip in patients with HF and secondary MR recently reported by the COAPT and MITRA-FR trials [32,33]. Several parameters such as excessively dilated LV and lower regurgitant volumes have been identified as potential factors that could justify the lack of benefit after MitraClip implantation in the MITRA-FR study [34]. Moreover, inclusion of patients with lower pulmonary pressure and better RV function in COAPT as compared to MITRA-FR trial, which did not exclude patients with severe RV dysfunction and PH, could also account for the diverging results between both trials. In this sense, we believe that assessment of TAPSE/PASP ratio, as a non-invasive index of the RV to PA coupling state, provides clinicians with a useful tool that might improve candidate selection for MitraClip among patients with secondary MR.

In addition, STS-Score, which assembles information on multiple variables with known prognostic relevance, also presented independent prognostic value in our sample, although weaker than that provided by TAPSE/PASP ratio. Integration of both STS-Score and TAPSE/PASP ratio may improve the assessment of a patients’ candidacy for TMVR with MitraClip. Of note, other known prognostic predictors in MitraClip candidates such as LVEF and LV volumes did not improve prognostic stratification in our study, but this should be regarded with caution as LVEF was only mildly reduced and LV volumes mild-to moderately enlarged across our sample. Similarly, severe TR, which was present in 11.8% of patients, also lacked prognostic value, possible due to lower LV volumes and higher LVEF values observed in these patients.

Notwithstanding, TAPSE/PASP ratio was unable to predict all-cause mortality on a separate basis, which could be due to the relatively low incidence of deaths and their frequent relation to non-cardiovascular causes. Accordingly, further trials with larger samples and follow-up periods that are powered for the detection of hard endpoints, as is mortality, are required to confirm our initial results.

### Limitations

Our study has several limitations. First, it is an observational study based on a retrospective analysis of a multicentre registry. Limitations inherent to this design as is the potential existence of confounding factors or selection and follow-up bias cannot be discarded. Moreover, we lacked a central core lab to assess imaging and procedural data and event adjudication were not crosschecked. Accordingly, definite conclusions regarding the clinical value of TAPSE/PASP ratio for the prediction of cardiovascular adverse events cannot be established from this single study and further adequately powered trials will be needed to confirm our initial results. Second, RHC was only conducted in a small subset of the total sample, at the discretion of the patient´s physician. This strategy represents real life practice an indeed, a systematic invasive assessment of right chambers pressures via RHC was not performed in the vast majority of studies evaluating prognostic predictors in patients undergoing MitraClip procedure. Thus, a stronger influence of hemodynamic parameters on outcomes cannot be excluded based on our results. Notwithstanding, non-invasive parameters employed in our study such as TAPSE/PASP ratio or echo derived PASP estimation have previously been validated against invasively assessed gold standards [24,26,29] and echo derived PASP displayed a significant correlation with invasively determined PASP in our sample, r = 0.592, *p* < 0.0001.

## 5. Conclusions

In this sample of HF patients with significant MR undergoing successful TMVR with MitraClip System, the TAPSE/PASP ratio emerged as a relevant prognostic predictor of the primary combined endpoint encompassing HF readmissions and all-cause mortality, regardless of MR aetiology and TR severity. Importantly, TAPSE/PASP ratio outperformed the prognostic value of each of its components, namely TAPS and PASP, on a separate basis. In agreement with prior investigations, TAPSE/PASP ratio displayed a significant correlation with other parameters evaluating the RV to PA coupling state such as PA compliance and PA pulse pressure, which reinforces its role as a non-invasive indicator of the right-sided cardiopulmonary unit.

## Figures and Tables

**Figure 1 jcm-10-01006-f001:**
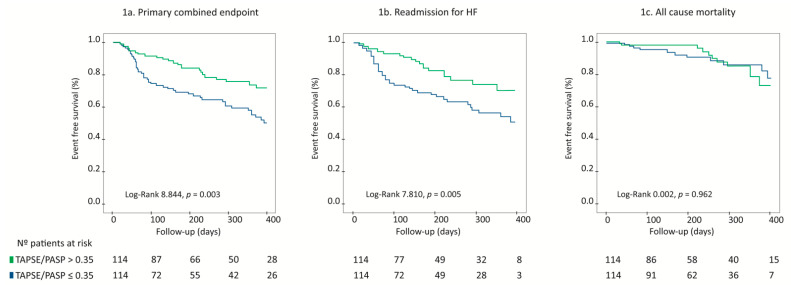
Survival analysis according to TAPSE/PASP ratio as shown by Kaplan–Meier curves. TAPSE/PASP ratio ≤ 0.35 is associated with higher rates of the primary combined endpoint (**a**), higher HF readmissions rates (**b**) without differences in all-cause mortality rates (**c**). TAPSE: tricuspid annular plane systolic excursion. PASP: pulmonary artery systolic pressure.

**Figure 2 jcm-10-01006-f002:**
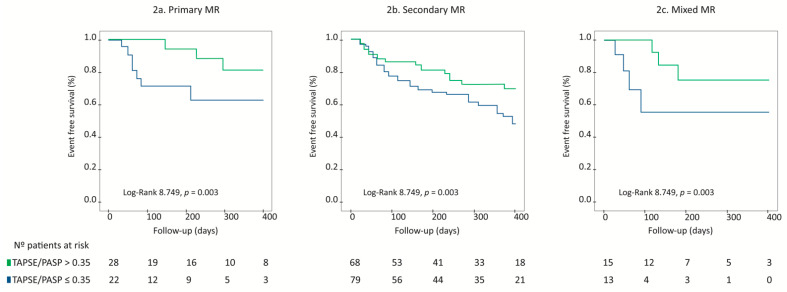
Survival analysis according to TAPSE/PASP ratio as shown by Kaplan–Meier curves, stratified according to MR aetiology. TAPSE/PASP ratio ≤ 0.35 is associated with higher rates of the primary combined endpoint in patients with primary MR (**a**), secondary MR (**b**) and mixed MR (**c**).

**Table 1 jcm-10-01006-t001:** Baseline characteristics according to TAPSE/PASP.

	TAPSE/PASP ≤ 0.35 (*n* = 114)	TAPSE/PASP > 0.35 (*n* = 114)	Total Sample (*n* = 228)	*p*-Value
**Clinical Characteristics**				
Age (years)	71.8 ± 11.8	73.3 ± 11.2	72.5 ± 11.5	0.327
Sex, male (*n*, %)	81(71.1)	73(64)	154(67.5)	0.258
Hypertension (*n*, %)	79(69.3)	86(76.1)	165(72.7)	0.250
Diabetes mellitus (*n*, %)	40(35.1)	34(29.8)	74(32.5)	0.396
Atrial fibrillation (*n*, %)	68(59.6)	65(58)	133(58.8)	0.893
Glomerular filtration rate (mL/min/1.73 m^2^)	58.2 ± 24.8	64 ± 26.1	60.9 ± 25.4	0.164
Stage 3b-5 chronic renal failure (*n*, %)	42(36.8)	33(28.9)	75(32.9)	0.205
Ischemic heart disease (*n*, %)	61(53.5)	52(45.6)	113(49.6)	0.400
Prior PCI (*n*, %)	41(36)	35(30.7)	76(33.3)	0.699
Prior CABG (*n*, %)	25(21.9)	17(14.9)	42(18.4)	0.156
Extra cardiac arteriopathy (*n*, %)	16(14)	19(16.7)	35(15.4)	0.512
Previous cardiac surgery (*n*, %)	38(33.3)	24(21.1)	62(27.2)	0.113
COPD (*n*, %)	25(21.9)	24(21.1)	49(21.5)	0.872
FC NYHA III-IV/IV (*n*, %)	101(88.6)	97(85.1)	198(86.8)	0.433
NT-proBNP	4821.6 ± 6364	4715.2 ± 5715	4762.2 ± 5990	0.915
EuroScore II	8.7 ± 8.3	8.2 ± 7.6	8.5 ± 7.9	0.589
STS-Score	5.4 ± 5.2	6.3 ± 5.5	5.8 ± 5.3	0.244
Medical therapy (*n*, %): Betablockers Mineralocorticoid antagonists ACE inhibitors/ARB/ARNis Diuretics	86 (76.8) 72 (64.3) 86 (76.8) 108 (96.4)	90 (78.9) 60 (52.6) 84 (73.7) 109 (95.6)	176 (77.9) 132 (58.4) 170 (75.2) 217 (96)	0.695 0.076 0.589 0.754
**Baseline Echocardiographic Characteristics**				
LVEDD (mm)	60.7 ± 9.9	61 ± 10.4	60.9 ± 10.1	0.838
LVESD (mm)	49.9 ± 12.5	49.9 ± 12.6	49.9 ± 12.5	0.975
Indexed LVEDV (mL/m^2^)	93.4 ± 34.3	90.9 ± 34.6	92.1 ± 34.4	0.623
Indexed LVEDV (mL/m^2^)	56.6 ± 31	55.2±31.3	55.8±31.1	0.762
LVEF (%)	39.3 ± 14.9	41.4 ± 15.5	40.4 ± 15.3	0.283
Indexed LA volume (mL/m^2^)	58.3 ± 21.2	62.3 ± 26.8	60.3 ± 24.2	0.291
MR grade IV/IV (*n*, %)	94(82.5)	98(86.7)	187(82.1)	0.373
Regurgitant Volume (mL)	60 ± 24.9	58.8 ± 19.2	59.5 ± 22.3	0.828
Effective Regurgitant Orifice (mm^2^)	42.6 ± 16.9	43.6 ± 18.5	43.1 ± 17.6	0.741
Indexed LVEDV/ Effective Regurgitant Orifice (mL/mm^2^)	2.5 ± 1.4	2.5 ± 1.3	2.5 ± 1.4	0.979
MR aetiology (*n*, %) * Primary Secondary Mixed	22(19.3) 79(69.3) 13(11.4)	28(25.2) 68(61.3) 15(13.5)	50(22.2) 147(65.3) 28(12.4)	0.439
Tricuspid regurgitation (*n*, %) Grade 0–1 Grade 2 Grade 3 Grade 4	35(30.7) 29(25.4) 29(25.4) 21(18.4)	63(55.3) 28(24.6) 17(14.9) 6(5.3)	98(43) 57(25) 46(20.2) 27(11.8)	<0.001
TAPSE (mm)	14.4 ± 3.3	19.5 ± 3.7	16.9 ± 4.3	<0.001
PASP (mmHg)	57.7 ± 11.9	40.2 ± 11.1	48.9 ± 14.4	<0.001
TAPSE/ PASP ratio	0.26 ± 0.1	0.52 ± 0.16	0.39 ± 0.18	<0.001
**Right Heart Catheterization ^ϕ^**				
Right atrial pressure (mmHg)	9.8 ± 4.8	8.5 ± 6.1	9.1 ± 5.5	0.520
LA mean pressure (mmHg)	25 ± 25.5	17.4 ± 7.6	20.7 ± 18	0.116
LA V-wave pressure (mmHg)	42.9 ± 13.8	29.8 ± 15.5	36 ± 15.9	0.010
Change in LA mean pressure (mmHg)	6.2 ± 6.6	5.8 ± 7.4	5.9 ± 7.1	0.848
Change in LA v-wave pressure (mmHg)	21.6 ± 15.7	12.8 ± 12.9	16.4 ± 14.6	0.081
PCW pressure (mmHg)	18.4 ± 7	19 ± 7.9	18.7 ± 7.3	0.833
Mean PA pressure (mmHg)	30.7 ± 12.4	29.5 ± 10.9	30.2 ± 11.5	0.795
Systolic PA pressure (mmHg) ^ϒ^	52.7 ± 18.9	41.1 ± 13.9	47.1 ± 17.4	0.016
PA Pulse Pressure (mmHg)	28.6 ± 13.6	23.2 ± 6.8	26 ± 11	0.214
Cardiac index (mL/min/m^2^)	2.3 ± 0.6	2.1 ± 0.6	2.2 ± 0.6	0.342
Pulmonary vascular resistance (WU)	2.9 ± 1.5	3.2 ± 1.2	3.1 ± 1.3	0.651
Transpulmonary pressure gradient (mmHg)	12.6 ± 7.6	10.8 ± 3.7	11.8 ± 6.1	0.442
Diastolic pressure gradient (mmHg)	2.9 ± 3.7	2.6 ± 2.5	2.7 ± 3.2	0.847
PA compliance (mL/mmHg)	2.4 ± 1.2	2.5 ± 1.3	2.4 ± 1.2	0.756

* values for 225 patients. **^ϕ^** values for 26 patients. ^ϒ^ values for 51 patients. ACE: angiotensin converting enzyme. ARB: angiotensin receptor blocker. ARNi: Angiotensin receptor neprilysin inhibitors. CABG: coronary artery bypass graft. COPD: chronic obstructive pulmonary disease. LA: left atrial. LVEDD: left ventricular end-diastolic diameter. LVEDV: left ventricular end-diastolic volume. LVEF: left ventricular ejection fraction. LVESD: left ventricular end-systolic diameter. LVESV: left ventricular end-systolic volume. MR: mitral regurgitation. PA: pulmonary artery. PASP: pulmonary artery systolic pressure. PCI: percutaneous coronary intervention. PCW: pulmonary capillary wedge. STS-Score: Society of Thoracic Surgery Score. TAPSE: tricuspid annular plane systolic excursion. WU: wood unit.

**Table 2 jcm-10-01006-t002:** Linear correlation analysis between TAPSE/PASP ratio and RHC-derived parameters.

RHC-Derived Parameters	Pearson/Spearman Correlation Coefficients	*p*-Value
Mean right atrial pressure (mmHg)	−0.102	0.606
Mean left atrial pressure (mmHg)	−0.197	0.142
Mean left atrial V-wave (mmHg)	−0.324	0.047
Mean PCW (mmHg)	−0.203	0.299
Mean PA Pressure (mmHg)	−0.236	0.246
Systolic PA Pressure (mmHg)	−0.373	0.007
PA Pulse Pressure (mmHg)	−0.423	0.028
Cardiac Index (mL/min/m^2^)	−0.003	0.988
Pulmonary vascular resistance (WU)	−0.103	0.615
Transpulmonary pressure gradient (mmHg)	−0.168	0.412
Diastolic pressure gradient (mmHg)	0.067	0.742
PA compliance (mL/mmHg)	0.418	0.030

TAPSE: tricuspid annular plane systolic excursion. PASP: pulmonary artery systolic pressure. PA: Pulmonary artery. PCW: Pulmonary capillary wedge pressure. RHC: right heart catheterization. WU: wood units.

**Table 3 jcm-10-01006-t003:** Multivariate Cox Regression Analysis.

Variables	β	Hazard Ratio	95% CI	*p*-Value
Diabetes Mellitus	0.442	1.56	0.92–2.64	0.101
NYHA FC III-IV/IV	0.722	2.06	0.73–5.84	0.175
TAPSE/PASP ≤ 0.35	0.693	2.0	1.13–3.53	0.017
STS-Score	0.050	1.05	1.01–1.10	0.024
TR grade III-IV/IV	0.091	1.09	0.62–1.92	0.750
LVEF	−0.003	0.99	0.98–1.01	0.765
**Regression Model Tests**				
Concordance = 0.662 (SE = 0.036)				
Likelihood ratio test				0.001
Wald test				0.002
Score (log rank) test				0.002
**Proportional Hazard Assumption**				
Diabetes Mellitus				0.54
NYHA FC				0.76
TAPSE/PASP ≤ 0.35				0.93
STS Score				0.96
TR				0.26
LVEF				0.54
Global				0.86

FC: functional class. NYHA: New-York heart association. LVEF: left ventricular ejection fraction. PASP: pulmonary artery systolic pressure. TAPSE: tricuspid annular plane excursion. TR: tricuspid regurgitation. SE: standard error. STS-Score: Society of Thoracic Surgery Score.

## Data Availability

The data presented in this study in contained within the article and Appendix A. Further data are available on request from the corresponding author.

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
