# Peer review of "Prognostic Role of TAPSE to PASP Ratio in Patients Undergoing MitraClip Procedure"

_jcm, 2021, doi:10.3390/jcm10051006_

Round 1

Reviewer 1 Report

The study "Prognostic role of TASPE to PASP ratio in patients undergoing MitraClip procedure" investigates the prognostic value of TAPSE/PASP concerning heart failure readmissions and all-cause mortality after transcatheter mitral valve repair.

The background and the study procedures are complicated to follow as many abbreviations are used, and some are not even explained in the text. Furthermore, the scientific approach and the study's objectives are not introduced properly, which makes it difficult to follow the presentation and the discussion of the results. Therefore, a fundamental revision of the manuscript is needed to be acceptable for publication in J Clin Med.

Section specific comments

Abstract

The abstract provides too many details about the outcome of the study (results). On the other hand, it lacks some important general information about the background and the objectives of the study. Furthermore, it does not become clear why TAPSE/PASP was compared to other parameters and what is meant by "the prognostic value of each of its components separately"?

L59: What is meant by independent predictor? Independent from which parameters?

Introduction

The introduction is very short, and it does not become clear how prognosis and risk assessment was done before based on accepted measures. Furthermore, the authors mentioned that the prognostic stratification in patients might be improved with the application of TAPSE/PASP ratio. However, no limitations or issues are introduced about the risk assessment based on other parameters. This additional information would explain why applying a novel approach based on the TASPE/PASP ratio would be beneficial.

Materials and Methods

Methods and data analysis is described very briefly. It would be beneficial to get more information about patient monitoring's clinical procedures (pre-, intra- and postoperative assessments) and how the data contributes to the diagnosis and assessments concerning transcatheter mitral valve repair.

The authors provided no rationale for dividing the sample into two groups based on the median value of TAPSE/PAPS ratio of 0.35 observed for the patient sample included in the study. As this threshold differs depending on the patient sample included, their generalization of the findings observed for these comparisons is highly questionable.

L109: was TAPSE measured for consecutive heartbeats as for atrial fibrillation patients? The same information is missing for other heartbeat related parameters as well.

L143: which parameters are considered as key variables? No variable was found to have P = 0.10 with Cox regression analysis. Which independed variable was selected for Cox regression?

Results

This section is difficult to read; it includes abbreviations without explanation and provides huge data tables without indications of how it contributes to the study's primary objective. Furthermore, the rationale for doing some of the analysis does not become apparent. The figures presented in the manuscript are just not readable.

L175: text from the manuscript template?

L188: what is meant by "developed the primary endpoint"?

Discussion

In the discussion section, the conclusions made should be referred more clearly to the study's specific outcomes and results. For example, it does not become clear based on which data the authors conclude that TAPSE/PASP provides a better prognostic stratification (L261). It should be discussed more in detail how the ratio is related to the outcome, what kind of preliminary clinical issues can be solved using this ratio and how the prognostic value could further be improved. A clear, structured argumentation is missing.

Reviewer 2 Report

In the manuscript “Prognostic role of TAPSE to PASP ratio in patients undergoing MitraClip procedure”, the authors aimed to investigate prognostic value of TAPSE to PASP ratio in patients undergoing transcatheter mitral valve repair. The topic of the manuscript is certainly interesting and captures one of the issues of the moment on cardiovascular outcomes in patients underwent MitraClip.

They found that TAPSE/PASP ratio emerges as a relevant prognostic predictor for the primary combined endpoint in patients undergoing successful TMVR with MitraClip. The analyses were appropriate and the manuscript is well written.

However, there are some points that need further clarification:

  1. There appear to be several errors in the text regarding percentages. In the same paragraph 147/228 it is reported first as 65.3% and then as 64.5%. Please double-check all percentages in the text and tables.

  1. There are no data regarding therapies. Pharmacologic treatment is critical to the prognosis of these patients. Please report these details.

  1. Please, specify more clearly whether the TAPSE/PASP ratio is before or after the MitraClip.

  1. What applications do the authors suggest in clinical practice and how do they think this may change the treatment approach in patients with severe mitral regurgitation.

  1. Please, you should explain each of your abbreviations the first time it appears in the main text(i.e. MR, ect.).

Author Response

Comments and Suggestions for Authors

In the manuscript “Prognostic role of TAPSE to PASP ratio in patients undergoing MitraClip procedure”, the authors aimed to investigate prognostic value of TAPSE to PASP ratio in patients undergoing transcatheter mitral valve repair. The topic of the manuscript is certainly interesting and captures one of the issues of the moment on cardiovascular outcomes in patients underwent MitraClip.

They found that TAPSE/PASP ratio emerges as a relevant prognostic predictor for the primary combined endpoint in patients undergoing successful TMVR with MitraClip. The analyses were appropriate, and the manuscript is well written.

However, there are some points that need further clarification:

1. There appear to be several errors in the text regarding percentages. In the same paragraph 147/228 it is reported first as 65.3% and then as 64.5%. Please double-check all percentages in the text and tables.

- Thank you for your important consideration. The mechanism of mitral regurgitation was not reported in three cases, leading to discordant values as the percentage of patients with secondary MR was referred to 225 patients. We have specified this in the text so that no misinterpretation occurs.

2. There are no data regarding therapies. Pharmacologic treatment is critical to the prognosis of these patients. Please report these details.

-Thank you for your important comment. Indeed, Guideline-directed medical therapy was optimized in all cases before they were considered for Mitraclip. We have added this information in Tables 1 and Supplementary Table 1.

3. Please, specify more clearly whether the TAPSE/PASP ratio is before or after the MitraClip.

- Certainly, this is an important consideration. We have specified in the text that TAPSE/PASP ratio investigated for prognostic purposes was measured at baseline, before MitraClip implantation.

In addition, TAPSE/PASP ratio after the procedure was also measured and was associated with post-procedural functional class.

4. What applications do the authors suggest in clinical practice and how do they think this may change the treatment approach in patients with severe mitral regurgitation.

- We believe that tricuspid annular plane systolic excursion (TAPSE) to PA systolic pressure (PASP) ratio contributes to improving the accuracy of pre-procedural risk assessment before transcatheter mitral valve repair (TMVR) with MitraClip. Accordingly, TAPSE/PASP should be considered together with other parameters that have proven to influence outcomes after TMVR, when determining the probable clinical benefit in a given candidate. Indeed, a low TAPSE/PASP ratio alone does not provide sufficient clinical basis to discard a patient for TMVR with MitraClip but does point out at an incipient decompensation of the right ventricle (RV) to its load, which is indicative of a higher risk situation. Accordingly, patients with a lower TAPSE/PASP ratio denoting RV to pulmonary arterial (PA) uncoupling that might be accepted for MitraClip could benefit from a more intensive post-procedural follow-up so that risk of subsequent heart failure (HF) readmissions and mortality is reduced.

In addition, we believe that TAPSE/PASP ratio may be especially relevant in patients with severe mitral regurgitation, as this condition is often associated with RV dysfunction and/or pulmonary hypertension (PH) at baseline. Although both RV dysfunction and PH entail a worse prognosis after MitraClip implantation, this intervention may still provide clinical benefits in selected candidates, as supported by prior studies. In this setting, a greater TAPSE/PASP ratio can help identify those patients that will benefit the most from TMVR.

Finally, lower TAPSE/PASP values, both at baseline and also when reassessed during follow-up, are associated with a worse functional class status. Therefore, TAPSE/PASP ratio can be employed to predict quality of life after MitraClip and over long-term follow-up.

Overall, TAPSE/PASP ratio appears as a relatively simple, easily available, non-invasive prognostic predictor, that might improve the assessment of the right heart to pulmonary unit, which has recently emerged as a novel, relevant prognostic predictor in other heart failure populations.

5. Please, you should explain each of your abbreviations the first time it appears in the main text (i.e., MR, etc.).

- Thank you for your comment. We have described each of the abbreviations that lacked an explanation in the text (i.e.: MR, HF, RV, BARC, LA).

Reviewer 3 Report

Thank you for allowing me to review your manuscript. This is very interesting. 

One question. Does preope or postope Moderate or greater TR affect outcomes? According to your results, no difference in TR between the groups was observed. However, TR is often a problem during the mitral surgery. 

Author Response

Thank you for important consideration. In our sample, TR was moderate or greater in 73 (32.0%) patients at baseline and in 52 (22.8%) after MitraClip intervention. Taking into account the prognostic significance exhibited by TR in prior studies, we have included moderate to severe TR on Cox regression analysis. However, moderate to severe TR exhibited no prognostic impact in patients undergoing TMVR in our sample. A possible explanation behind this finding could be the association of moderate to severe TR with lower left ventricular volumes as well as higher LVEF.  

Round 2

Reviewer 1 Report

Thanks for considering my comments and revising the manuscript accordingly. The most critical aspect was to provide additional information about setting the threshold for TAPSE/PASP to 0.35. This is now explained in one sentence (L187f) and could be elaborated more in detail as provided in the author's response. Furthermore, you refer to the median value of TAPSE/PASP to be 0.35 rather than TAPSE (as written in L187)?

Some other minor points include the missing explanation of HF in the main text (L80) and avoiding the strange wording "Bleeding Academic Research Consortium (BARC) bleeding" to "bleeding assessed based on Bleeding Academic Research Consortium (BARC)", or similar.

After these minor revisions, the manuscript can be accepted for publication.

Author Response

Thanks for considering my comments and revising the manuscript accordingly. The most critical aspect was to provide additional information about setting the threshold for TAPSE/PASP to 0.35. This is now explained in one sentence (L187f) and could be elaborated more in detail as provided in the author's response. Furthermore, you refer to the median value of TAPSE/PASP to be 0.35 rather than TAPSE (as written in L187)?

- Thank you for your comment. We have elaborated on the cutoff value of 0.35 for the TAPSE/PASP ratio in the manuscript and have also corrected the sentence in L187 which contained an erratum, as 0.35 represents the median value of the TAPSE/PAPS ratio.

Some other minor points include the missing explanation of HF in the main text (L80) and avoiding the strange wording "Bleeding Academic Research Consortium (BARC) bleeding" to "bleeding assessed based on Bleeding Academic Research Consortium (BARC)", or similar.

- Thank you for your considerations. We have provided an explanation for HF in Line 80 in the text and have modified the wording in the sentence describing the employed bleeding classification, which now stands as “bleeding assessed according to Bleeding Academic Research Consortium (BARC) criteria”.